The increasing prevalence of HIV/Helicobacter pylori co-infection over time, along with the evolution of antiretroviral therapy (ART)

Radovanović Spurnić Aleksandra spurnic@yahoo.com 1
Brmbolić Branko 1
Stojšić Zorica 2
Pekmezović Tatijana 3
Bukumirić Zoran 4
Korać Miloš 1
Salemović Dubravka 1
Pešić-Pavlović Ivana 5
Stevanović Goran 1
Milošević Ivana 1
Jevtović Djordje 1
1 Hospital for Infectious and Tropical Diseases, Clinical Centre of Serbia, Faculty of Medicine, University of Belgrade , Belgrade , Serbia
2 Institute of Pathology, Faculty of Medicine, University of Belgrade , Belgrade , Serbia
3 Institute of Epidemiology, Faculty of Medicine, University of Belgrade , Belgrade , Serbia
4 Institute of Medical Statistics and Informatics, Faculty of Medicine, University of Belgrade , Belgrade , Serbia
5 Virology Laboratory, Microbiology Department, Clinical Center of Serbia , Belgrade , Serbia
Hill-Cawthorne Grant
Electronic publication date: 2017 May 30
Publication date: 2017
Volume: 5
Electronic Location ID: e3392
Received 2017 Feb 8; Accepted 2017 May 9
Copyright: ©2017 Radovanović Spurnić et al.
Copyright year: 2017
Copyright holder: Radovanović Spurnić et al.
License: This is an open access article distributed under the terms of the Creative Commons Attribution License, which permits unrestricted use, distribution, reproduction and adaptation in any medium and for any purpose provided that it is properly attributed. For attribution, the original author(s), title, publication source (PeerJ) and either DOI or URL of the article must be cited.
License URL: https://creativecommons.org/licenses/by/4.0/

Keywords: Helicobacter pylori, ART, HIV, Gastritis histology, Esophagogastroduodenoscopy

Funding: Ministry of Education, Science and Technological Development of the Republic of Serbia 175081 The work was supported by grant 175081 from the Ministry of Education, Science and Technological Development of the Republic of Serbia. The funders had no role in study design, data collection and analysis, decision to publish, or preparation of the manuscript.

==============================
Helicobacter pylori (H. pylori) is one of the most common human bacterial infections with prevalence rates between 10–80% depending upon geographical location, age and socioeconomic status. H. pylori is commonly found in patients complaining of dyspepsia and is a common cause of gastritis. During the course of their infection, people living with HIV (PLHIV) often have a variety of gastrointestinal symptoms including dyspepsia and while previous studies have reported HIV and H. pylori co-infection, there has been little data clarifying the factors influencing this. The aim of this case-control study was to document the prevalence of H. pylori co-infection within the HIV community as well as to describe endoscopic findings, gastritis topography and histology, along with patient demographic characteristics across three different periods of time during which antiretroviral therapy (ART) has evolved, from pre- highly active antiretroviral therapy (HAART) to early and modern HAART eras. These data were compared to well-matched HIV negative controls. Two hundred and twelve PLHIV were compared with 1,617 controls who underwent their first esophagogastroduodenoscopy (EGD) to investigate dyspepsia. The prevalence of H. pylori co-infection among PLHIV was significantly higher in the early (30.2%) and modern HAART period (34.4%) compared with those with coinfection from the pre-HAART period (18.2%). The higher rates seen in patients from the HAART eras were similar to those observed among HIV negative controls (38.5%). This prevalence increase among co-infected patients was in contrast to the fall in prevalence observed among controls, from 60.7% in the early period to 52.9% in the second observed period. The three PLHIV co-infected subgroups differed regarding gastritis topography, morphology and pathology. This study suggests that ART has an important impact on the endoscopic and histological features of gastritis among HIV/H. pylori co-infected individuals, raising the possibility that H. pylori-induced gastritis could be an immune restoration disease.

Introduction

Thirty years after discovery, the global HIV/AIDS pandemic is still ongoing. Approximately 78 million people have been affected to date, with more than 36 million currently living with HIV infection or AIDS (UNAIDS, 2016).

During the course of the disease, people living with HIV (PLHIV) often have a variety of gastrointestinal symptoms including dyspepsia. Gastrointestinal symptoms can be caused by HIV per se, opportunistic and non-opportunistic infections including Helicobacter pylori (H. pylori), and adverse effects of highly active antiretroviral therapy (HAART) (Serlin & Dieterich, 2008; Edwards et al., 1991; Nkuize et al., 2010).

H. pylori has the main role in the pathogenesis of chronic gastritis, peptic ulcer disease, gastric cancer, MALT lymphoma and several extra-gastric manifestations (Kusters, Van Vliet & Kuipers, 2006; Wotherspoon et al., 1991). It is one of the most common bacterial pathogens affecting the general population (Logan & Walker, 2001) and its prevalence is estimated to be up to 50% worldwide (Höcker & Hohenberger, 2003). The risk factors and routes of transmission still remain unclear. The most important risk factor seems to be a lower socioeconomic status in early childhood, allowing easy person-to-person transmission (Brown, 2000; Stone, 1999).

The prevalence of H. pylori infection among PLHIV varies from 10 to 76%, depending on the time period, geographical location and population (Nkuize et al., 2010; Marano, Smith & Bonanno, 1993; Fialho et al., 2011; Chiu et al., 2004; Mohamed et al., 2002). Earlier studies reported lower prevalence of this co-infection among PLHIV compared to matched HIV negative controls, contrary to data from the recent HAART era, where the H. pylori prevalence rate in PLHIV approaches the rate observed in general population.

Material and Methods

The aim of this study was to analyze the prevalence H. pylori co-infections, within the HIV community as well as to describe endoscopic findings, gastritis topography and histology, along with patient demographic characteristics across three different periods of time, during which antiretroviral therapy (ART) has evolved, from pre-HAART (1993–1997), to the early HAART (1999–2003) and modern HAART era (2011–2015).

This retrospective case-control study was carried out at the Hospital for Infectious and Tropical Disease of the Clinical Center of Serbia at the Faculty of Medicine, University of Belgrade, which is our the major reference center for the management of HIV infections. The study included HIV positive patients who underwent esophagogastroduodenoscopy (EGD) due to dyspeptic symptoms.

Study population

For the purpose of this study, patients from the PLHIV cohort were stratified into three categories based on the time of EGD performance: the first group (G1) consisted of patients from the last five years of the pre-HAART period, who underwent EGD between January 1st 1993 and December 31st 1997 (the pre-HAART group). The second group (G2) included patients who underwent EGD between January 1st 1999 and December 31st 2003, the first five years of HAART (the early HAART group). The third group (G3) included patients who underwent EGD between January 1st 2011 and December 31st 2015 (the modern HAART group). As HAART became available in Serbia in 1998, patients who underwent EGD from January 1st 1998 to December 31st 1998 were excluded from study in order to preserve the homogeneity of the groups.

The H. pylori/HIV co-infection prevalence and demographic characteristics across three groups were compared with the same parameters of HIV-negative controls who underwent EGD during the same three periods of time. Controls were matched with the study population.

Each patient/control was included in one HAART period group only.

The study inclusion and exclusion criteria

The study inclusion criteria were: dyspepsia (including epigastric discomfort, nausea, vomiting, heartburn, burping, loss of body weight, bloating and stomach cramps) lasting up to three months before their first EGD (after informed consent), with gastric biopsy and histology.

The study exclusion criteria were: gastrointestinal bleeding, odynophagia, dysphagia, chest pain, chronic diarrhea (which could have jeopardized results) and being under 18 years of age.

Data collection and instruments

Demographic characteristics (age, gender, education level), risk factors for HIV acquisition, duration of HIV, CD4 cell counts, HIV viral loads and ART data were collected from the hospital database. The CD4 cell counts and HIV viral load were accepted as appropriate if performed within one month before or after EGD. Endoscopic macroscopic finding were collected from the hospital database. Histological data were collected from the database of the Institute of Pathology at the Faculty of Medicine, University of Belgrade.

The HIV infection was assessed using commercial immunoassays in accordance with the manufacturer protocols. The CD4 cells were quantified by flow cytometry. Plasma HIV-1 RNA loads were measured by a quantitative reverse transcriptase polymerase chain reaction (Ultrasensitive assay version 1.5, Roche Molecular Systems, Branchburg, NJ, USA), with a lower detection limit of 20 copies/mL (1.3 log10).

Endoscopy and histology

EGD was performed in the standard manner (the conventional oral route) with Olympus endoscope, after overnight fasting of approximately 10 h. Four biopsy specimens (two from the gastric body and two from the antrum) were taken using the standard biopsy forceps from each patient. Standard terminology was used for the endoscopic diagnosis. Biopsy samples were fixed in formalin, embedded in paraffin, cut into 4-µm sections, and stained by hematoxylin and eosin (H&E) and modified Giemsa stain. The histological appearance of gastric biopsy was assessed as either normal or gastritis. The gastritis was classified according to the updated Sydney System (Dixon et al., 1996).

Statistics

All analyses were performed using an electronic database organized in the IBM SPSS Statistics 22 (SPSS Inc., Chicago, IL, USA) statistical package. Patient baseline characteristics were analyzed using descriptive statistical methods, while non-parametric variables were analyzed using One-way ANOVA with post-hoc Tukey HSD Test, Kruskal–Wallis test, Mann–Whitney test, Chi-square test for trend and Fisher’s exact test, as appropriate. The level of significance was 0.05.

Study approval

The survey protocol was conducted according to the Declaration of Helsinki. The protocol was approved by Ethics Committee of Clinical Centre of Serbia and granted by the Ministry of Education, Science and Technological Development of the Republic of Serbia (No. 175081). All survey participants signed informed consent forms before undergoing EGD.

Results

A total of 212 PLHIV and 1617 HIV negative patients, all with dyspeptic symptoms who underwent endoscopic examination during the three defined HAART periods were included in this retrospective case-control study. The three control groups, for the pre-HAART, early HAART and modern HAART periods consisted of 598, 603 and 416 patients, respectively.

The pre-HAART HIV positive group (G1) consisted of 66 patients, of which 44 (66.7%) were males while the remaining 22 (33.3%) were females, with a mean age of 38.0 ± 10.5 years. The youngest patient was aged 21, while the oldest patient was 68 years old. The prevalence of H. pylori infection in this group was 18.2% (Table 1, Fig. 1).

Table 1 Characteristics of people living with HIV (PLHIV) in three periods of antiretroviral therapy.

(Note: Significant P value (<0.05) are in bold). P values were calculated with One-way ANOVA with post-hoc Tukey HSD for normally distributed continuous variables, Kruskal–Wallis rank test and Mann Whitney test for not normally distributed continuous or ordinal variables, Pearson Chi-Square test or Fisher’s exact test for categorical variables.

	G1	G2	G3	P-value	
	n = 66	n = 53	n = 93	G1 vs. G2 vs. G3	G1 vs. G2	G1 vs. G3	G2 vs. G3	
Age (years), mean ± sd	38.0 ± 10.5	40.8 ± 10.6	42.2 ± 11.8	0.059	0.348	0.047	0.739	
Gender, n (%)	
Male	44 (66.7)	32 (60.4)	74 (79.6)	0.055	0.478	0.067	0.012	
Female	22 (33.3)	21 (39.6)	19 (20.4)					
Educational level, n (%)	
Primary	15 (28.3)	14 (29.2)	8 (8.6)	0.016	0.719	0.007	0.044	
Secondary	27 (50.9)	21 (43.8)	55 (59.1)					
Higher	11 (20.8)	13 (27.1)	30 (32.3)					
CD4 count/µL, median (range)	97 (4–545)	250 (6–792)	435 (2–1622)	<0.001	0.001	<0.001	<0.001	
CD4 count/µL >  200, n (%)	19 (35.2)	28 (57.1)	75 (80.6)	<0.001	0.025	<0.001	0.003	
The length of HIV infection, median (range)	1 (0–11)	2.5 (0–15)	5 (0–29)	<0.001	0.003	<0.001	0.008	
PCR cp/ml, median (range)	–	0 (0–1223393)	20 (0–10000000)	–	–	–	0.527	
HIV risk behavior, n (%)	
IVDU	30 (52.6)	11 (22.9)	13 (14.0)	<0.001	0.006	<0.001	0.101	
Sex	15 (26.3)	28 (58.3)	57 (61.3)	
Blood/Blood product	6 (10.5)	5 (10.4)	4 (4.3)	
Unknown	6 (10.5)	4 (8.3)	19 (20.4)					
AIDS, n (%)	45 (78.9)	40 (83.3)	62 (66.7)	0.063	0.569	0.106	0.036	
Notes.

G1 1993–1997

G2 1999–2003

G3 2011–2015

PCR Polymerase chain reaction

IVDU An Intravenous Drug User

AIDS Acquired Immune Deficiency Syndrome

SD standard deviation

n frequency

Figure 1 The prevalence of Helicobacter pylori co-infection in people living with HIV (PLHIV) in three different time points of antiretroviral therapy.

G1 = 1993–1997; G2 = 1999–2003; G3 = 2011–2015; HAART- the high active antiretroviral therapy G1 vs. G2 p = 0.125; G1 vs. G3 p = 0.024; G2 vs. G3 p = 0.602; G1 vs. G2 vs. G3 p = 0.028. P values were calculated using Pearson Chi-Square test. Significant P value (<0.05) are in bold.

The early-HAART HIV positive group (G2) consisted of 53 patients, of which 32 (60.4%) were males while 21 (39.6%) were females. The mean age was 40.8 + 10.6 years. The youngest patient was 18, while the oldest patient was 63 years old. The prevalence of H. pylori infection in this group was 30.2% (Table 1, Fig. 1).

The modern-HAART HIV positive group (G3) consisted of 93 patients, of whom 74 (79.6%) were males and 19 (20.4%) were females, with the mean age of 42.2 + 11.8 years. The youngest patient was 24, while the oldest patient was 83 years old. The prevalence of H. pylori infection in this group was 34.4 % (Table 1, Fig. 1).

The age and gender of patients, across all three groups did not differ significantly, although there were more males in G3 than in G2 (p = 0.012). In all three groups the majority of patients had secondary education; the level of education did not differ between G1 and G2 (p = 0.719), while G3 patients had somewhat higher level of education than G1 (p = 0.007) and G2 (p = 0.044). The median CD4 cell count/µL was significantly different across groups (97 count/µL (4–545), 250 count/µL (6–792) and 435 count/µL (2–1622), in groups G1–3, respectively, p < 0.001). G3 patients had the highest CD4 + counts, while the lowest count was observed in G1 patients. A significantly lower prevalence of AIDS was observed in G3 than in G2 patients. The level of HIV viremia did not differ significantly between G2 and G3. The level of HIV viremia had not been measured in the pre-HAART era (Table 1).

The median duration of HIV infection from initial diagnosis to the EGD was significantly longer in G3 (5 years (0–29)) compared to G2 (2.5 years (0–15)) and G1 (1 year (0–11)) (p < 0.001).

There were 30 (52.6%) intravenous drug users (IVDUs) in G1, while the sexual route of HIV transmission was more prevalent among patients in groups G2 and G3 (p < 0.001). There was no significant difference in the route of HIV transmission between G2 and G3 (p = 0.101) (Table 1).

The sexual HIV transmission routes included homosexual (MSM), bisexual and heterosexual PLHIV. MSM was the most prevalent in G3. In G1 and G2 however, heterosexuals prevailed (p < 0.001) (Table 2). Males prevailed across all groups (p < 0.001).

Table 2 Sexual transmission route of HIV infection was homosexual (MSM), heterosexual and bisexual.

	G1	G2	G3	P-value	
	n = 15	n = 28	n = 57	G1 vs. G2 vs. G3	G1 vs. G2	G1 vs. G3	G2 vs. G3	
Sexual transmission of HIV infection, n (%)				<0.001	0.898	0.004	<0.001	
Sex	9 (60.0)	19 (67.9)	31 (54.4)					
Bisex	3 (20.0)	5 (17.9)	0 (0)					
MSM	3 (20.0)	4 (14.3)	26 (45.6)					
Notes.

G1 1993–1997

G2 1999–2003

G3 2011–2015

n frequency

Significant P value (<0.05) are in bold.

P values were calculated using Fisher‘s exact test.

The number of PLHIV with pathological EGD findings (erythema, edema, visible vascular pattern, mucosal friability, erosive or hemorrhagic lesion, exudate) were significantly different across groups, with 49 (74.2%) patients in G1, 28 (52.8%) patients in G2 and 38 (40.9%) patients in G3 (G1 vs. G2 p = 0.015; G1 vs. G3 p < 0.001; G2 vs. G3 p = 0.162; G1 vs. G2 vs. G3 p < 0.001). More precisely, pathological EGD findings were more common in G1 than in either G2 or G3 patients. In the latter two groups the number of patients with pathological EGD findings was similar (Table 3).

Table 3 Characteristics of endoscopy and histological findings in people living with HIV (PLHIV) with Helicobacter pylori co-infection during three different periods (three different ARTs).

	G1	G2	G3	P-value	
	n = 12	n = 16	n = 32	G1 vs. G2 vs. G3	G1 vs. G2	G1 vs. G3	G2 vs. G3	
H. pylori infection positive, n (%)	12 (18.2)	16 (30.2)	32 (34.4)	0.028	0.125	0.024	0.602	
EGD pathological findings (macroscopically finding), n (%)	49 (74.2)	28 (52.8)	38 (40.9)	<0.001	0.015	<0.001	0.162	
Gastritis topographical localization, n (%)				<0.001	0.020	<0.001	0.051	
Antrum	11(91.7)	7 (43.8)	9 (28.1)					
Corpus	0 (0)	2 (12.5)	0 (0)					
Pan-gastritis	1 (8.3)	7 (43.8)	23 (71.9)					
H. pylori density, n (%)	
Mild (+)	5 (41.7)	10 (62.5)	5 (15.6)	<0.001	0.132	0.050	<0.001	
Moderate (+ +)	4 (33.3)	6 (37.5)	10 (31.3)					
Severe (+ +  +)	3 (25.0)	0 (0)	17 (53.1)					
Activity, n (%)	
No	2 (16.7)	1 (6.3)	0 (0)					
Mild (+)	8 (66.7)	10 (62.5)	16 (50.0)	0.055	0.314	0.026	0.157	
Moderate (+ +)	1 (8.3)	4 (25.0)	12 (37.5)					
Severe (+ +  +)	1 (8.3)	1 (6.3)	4 (12.5)					
Lymphoid follicles, n (%)	
No	9 (75.0)	14 (87.5)	21 (65.6)	0.246	0.401	0.497	0.102	
Mild (+)	3 (25.0)	2 (12.5)	9 (28.1)					
Moderate (+ +)	0 (0)	0 (0)	2 (6.3)					
Intestinal metaplasia, n (%)				0.107	1.000	0.405	0.238	
No	11(91.7)	15 (93.8)	24 (75.0)					
Mild (+)	1 (8.3)	1 (6.2)	8 (25.0)					
Notes.

G1 1993–1997

G2 1999–2003

G3 2011–2015

n frequency

Significant P value (<0.05) are in bold.

P values were calculated using Kruskal–Wallis test or Mann Whitney test for not normally distributed continuous or ordinal variables, (Fisher’s exact test for categorical variables) but not normally distributed.

In PLHIV, the gastritis topography significantly differed between H. pylori positive subgroups (p < 0.001). EGD revealed antral gastritis as the most common in G1 H. pylori positive subgroups patients. The majority of G2 H. pylori positive subgroups patients had either antral or pan-gastritis, while G3 H. pylori positive subgroups patients mostly suffered from pan-gastritis. The histopathological grading of gastritis activity was similar between subgroups, while H. pylori density differed significantly (p < 0.001). The most severe density was recorded in G3 H. pylori positive subgroups patients. Although gastritis activity was similar across subgroups, the mildest cases prevailed in G1, and in G3 there were more patients with moderate activity (p = 0.026). The intensity of lymphoid follicles, mononuclear cell infiltration, and the intestinal metaplasia were similar across subgroups. All cases of gastritis in the presented H. pylori positive cohort were classified as chronic non-atrophic gastritis. Intestinal metaplasia was only sporadically present, mostly replacing one to few foveolae (Table 3).

Patients in the G1 group with H. pylori co-infection were younger than in G2 and G3 subgroups (p = 0.018). Gender distribution was similar across H. pylori positive subgroups. The median CD4 cell counts were lower in G1 than in G3 H. pylori positive subgroup patients (p = 0.013) (Table 4).

Table 4 Characteristic of people living with HIV (PLHIV) with Helicobacter pylori co-infection.

	G1	G2	G3	P value	
	n = 12	n = 16	n = 32	G1 vs. G2 vs. G3	G1 vs. G2	G1 vs. G3	G2 vs. G3	
Age (years), mean ± sd	34.6 ± 7.1	44.4 ± 10.3	43.2 ± 10.0	0.018	0.026	0.027	0.918	
Gender, n (%)	
Male	5 (41.7)	12 (75.0)	24 (75.0)	0.061	0.121	0.071	1.000	
Female	7 (58.3)	4 (25.0)	8 (25.0)					
CD4 count/µL, median (range)	380 (230–545)	380 (210–792)	518.5 (35–1622)	0.043	0.739	0.013	0.144	
Notes.

G1 1993–1997

G2 1999–2003

G3 2011–2015

SD standard deviation

n frequency

Significant P value (<0.05) are in bold.

P values were calculated using One-way ANOVA with post-hoc Tukey HSD for normally distributed continuous variables, Kruskal–Wallis rank and Mann Whitney test for not normally distributed continuous or ordinal variables, Fisher’s exact test for categorical variables.

The age of the controls differed significantly between three the HAART periods. Control patients from the modern HAART were significantly older than control patients from the pre-HAART and early HAART periods (p < 0.001) (Table 5).

Table 5 Ages of HIV negative controls and prevalence of Helicobacter pylori infection in three different observed periods.

	G1	G2	G3	P-value	
	n = 598	n = 603	n = 416	G1 vs. G2 vs. G3	G1 vs. G2	G1 vs. G3	G2 vs. G3	
Age (years), mean ± sd	44.5 ± 14.3	45.3 ± 14.2	48.8 ± 14.9	<0.001	0.576	<0.001	<0.001	
H. pylori positive, n (%)	363 (60.7)	319 (52.9)	160 (38.5)	<0.001	0.006	<0.001	<0.001	
Notes.

G1 1993–1997

G2 1999–2003

G3 2011–2015

SD standard deviation

n frequency

Significant P value (<0.05) are in bold.

P values were calculated using One-way ANOVA with post-hoc Tukey HSD for normally distributed continuous variables and Pearson Chi-Square test for categorical variables.

The prevalence of H. pylori infection significantly varied between control groups, decreasing over time from 60.7% in pre-HAART to 38.5% in modern HAART era controls. In modern HAART controls, the prevalence of H. pylori infection was similar to the prevalence of H. pylori infection co-infected modern HAART individuals (Fig. 2).

Figure 2 The prevalence of Helicobacter pylori infection in people living with HIV (PLHIV) vs. HIV negative patients in three time points of antiretroviral therapy.

G1 = 1993–1997; G2 = 1999–2003; G3 = 2011–2015; HAART = the high active antiretroviral therapy. Significant P value (<0.05) are in bold. P values were calculated using Pearson Chi-Square test.

Discussion

From the very beginning of the HIV/AIDS epidemic in Serbia, it became evident that these patients suffered from various gastrointestinal symptoms, including dyspepsia. The cause of gastrointestinal symptoms could be HIV per se, opportunistic and non-opportunistic infections including Helicobacter pylori (H. pylori), along with adverse effects of highly active antiretroviral therapy (HAART). Therefore, EGD has become an important differential diagnostic tool in the evaluation of gastrointestinal disease in PLHIV (Corley, Cello & Koch, 1999; Korać et al., 2008). ART naïve PLHIV live with the impairment of cell-mediated immunity and are therefore at high risk for opportunistic infections and opportunistic tumors. Introduction of ART in 1986, which evolved from rather a weak mono and/or dual therapy to more potent HAART in 1996, has dramatically changed the course of HIV infection, converting it to a chronic, not necessarily a fatal disease. Good patient adherence to treatment and infection monitoring are of crucial importance (Dieffenbach & Fauci, 2011; Jevtović et al., 2007).

Occasionally, however, especially among late presenters in which HAART is initiated only at a point of profound immunodeficiency, the immune system recovery may be accompanied with the emergency of complications called the immune reconstitution inflammatory syndrome (IRIS). IRIS is a consequence of the restored immune response to specific infectious or non-infectious antigens (DeSimone, Pomerantz & Babinchak, 2000; French et al., 2000; Mayer et al., 2004; Shelburne, Montes & Hamill, 2006). According to our data, the prevalence of HIV/ H. pylori co-infection in Serbia ranges from 18.2% to 34.4%, depending on the study period, namely the type of applied ART. Indeed, the prevalence of HIV/H. pylori co-infection has increased over time, contrary to reduction in H. pylori infection incidence in HIV negative controls. A similar prevalence of H. pylori infection was observed among PLHIV and HIV negative controls in the modern HAART era (Fig. 2). We have not explored the causes of decreasing H. pylori infection prevalence in the HIV negative population, but a possible explanation is the H. pylori infection has had a decreasing trend, most probably by effective application of H. pylori eradication therapy (Malfertheiner et al., 2012).

It seems that HAART may be one of the predisposing factors in PLHIV. HAART enables the immune system recovery, which is associated with an increase in the CD4 cell counts to almost normal values. Altogether, this could play an important role in the pathogenesis of gastritis associated with HIV/H. pylori infection.

In accordance with published data (Nkuize et al., 2010; Marano, Smith & Bonanno, 1993; Fialho et al., 2011; Chiu et al., 2004; Mohamed et al., 2002; Olmos et al., 2004; Nkuize et al., 2012), we also demonstrated the highest CD4 cell counts in HIV/H. pylori co-infected patients. Moreover, CD4 cell counts show a positive increasing trend simultaneous with ART evolution from pre-HAART to modern HAART. This is in concordance with the rising prevalence of H. pylori induced gastritis among co-infected patients.

To the best of our knowledge, this is the first EGD-based morphological comparison of HIV/H. pylori co-infected patients in three different periods of ART evolution. We have demonstrated a significantly reduced incidence of pathological EGD findings in PLHIV with dyspeptic symptoms in the modern HAART era in comparison with PLHIV co-infected during the pre-HAART era. Then, in three time points H. pylori gastritis topography has also changed significantly between the three HAART periods. During the pre-HAART era, gastritis was characterized by mild H. pylori density and moderate activity. This is contrary to gastritis observed in the modern HAART group, where the density became mostly severe, while the activity remained mild. The changes in gastric histological findings may be associated with an increased CD4 cell count in H. pylori positive subgroup, and could suggest that the type and duration of immune and inflammatory responses are responsible for differences in histological presentations of H. pylori associated gastritis. The direct impact of ART on H. pylori infection and gastritis should also be taken into consideration, especially since there is evidence for ART-induced GI symptoms (Wilcox & Saag, 2008).

Finally, it is not only that HAART changed the course of HIV/H. pylori co- infection. Over time, the profile of patients suffering from HIV infection has changed. In Serbia, the first PLHIV were IVDU (Vucic-Jankovic, Ristic & Bakovic, 1996) and this was the most prevalent mode of HIV transmission during the entire pre-HAART period. The epidemiological situation in Serbia is now completely different. Today, in the modern HAART era, the patient population is mostly male, while the dominant route of HIV transmission is sexual intercourse, more precisely MSM. This is reflected in the gender distribution in the modern HAART PLHIV group. In contrast, no differences in gender prevalence were found in the three PLHIV H. pylori positive subgroups which corresponds to the findings of the HIV negative patients infected with H. pylori (Kusters, Van Vliet & Kuipers, 2006; Brown, 2000). Despite the differences in HIV risk behavior, the duration of HIV infection and educational level, PLHIV in the three HIV positive groups from different HAART eras are of similar age as opposed to the three PLHIV H. pylori positive subgroups. In the modern HAART era, PLHIV with H. pylori co-infection were significantly older. This finding is in accordance with the age trend among HIV negative controls.

In the modern HAART era, PLHIV are living longer, are normally integrated in society, with similar risks and diseases as HIV-negative individuals.

Study limitations

This study has several limitations. First, it is a retrospective case-control study. Second, various risk factors for H. pylori acquisition, such as exposure to antibiotics, as well as to proton-pump inhibitors (PPI) and H2 blocker therapies, along with hypochlorhydria, body mass index, particular habits, which have all been addressed by numerous studies before (Fialho et al., 2011; Chiu et al., 2004; Olmos et al., 2004; Nkuize et al., 2012), were not explored. Third, we have not sufficiently explored H. pylori acquisition in HIV negative cohorts. Despite these limitations, we believe that the study represents new research findings on the relationship between ART development over time, HIV risk factors, H. pylori infection, gastritis distribution and topography.

Conclusion

Our data may also suggest that H. pylori needs a functional immune system to induce human gastric mucosa inflammation. It is necessary for future studies to investigate the difference between epidemiological characteristics of H. pylori infection in the PLHIV and HIV negative population, expanding conclusions and implications from our study. Altogether, our findings suggest that HAART has an important impact on the EGD features of HIV/H. pylori co-infection.

Supplemental Information

Supplemental Information 1 STROBE checklist

Click here for additional data file.

Supplemental Information 2 Database

Click here for additional data file.

Supplemental Information 3 Raw data

Each worksheet contains the raw data used to generate one of the tables or figures in the manuscript.

Click here for additional data file.

Additional Information and Declarations

Competing Interests

Author Contributions

Human Ethics

Data Availability

The authors declare that they have no competing interests.

Aleksandra Radovanović Spurnić conceived and designed the experiments, analyzed the data, wrote the paper, prepared figures and/or tables.

Branko Brmbolić conceived and designed the experiments, performed the experiments, analyzed the data, wrote the paper, reviewed drafts of the paper.

Zorica Stojšić conceived and designed the experiments, performed the experiments, analyzed the data, contributed reagents/materials/analysis tools, wrote the paper, reviewed drafts of the paper.

Tatijana Pekmezović conceived and designed the experiments, wrote the paper.

Zoran Bukumirić conceived and designed the experiments, analyzed the data, contributed reagents/materials/analysis tools, prepared figures and/or tables.

Miloš Korać conceived and designed the experiments, performed the experiments, wrote the paper.

Dubravka Salemović wrote the paper.

Ivana Pešić-Pavlović performed the experiments, contributed reagents/materials/analysis tools.

Goran Stevanović and Ivana Milošević participated in implementation of the study, data collestion and database management.

Djordje Jevtović conceived and designed the experiments, analyzed the data, wrote the paper, reviewed drafts of the paper.

The following information was supplied relating to ethical approvals (i.e., approving body and any reference numbers):

The survey protocol was conducted according to the Declaration of Helsinki. The protocol was approved by the Clinical Centre of Serbia Ethics Committee and granted by Ministry of Education, Science and Technological Development of the Republic of Serbia (No. 175081). All survey participants signed informed consent forms before undergoing EGD.

The following information was supplied regarding data availability:

The raw data has been supplied as a Supplementary File.

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
