# Peer review of "The increasing prevalence of HIV/Helicobacter pylori co-infection over time, along with the evolution of antiretroviral therapy (ART)"

_PeerJ, doi:10.7717/peerj.3392_

## Round 0.1 · original submission · Minor Revisions

This is an interesting and important topic and we would be happy to accept it for publication in PeerJ after a minor revision. This revision will need to address the reviewers’ comments as well as be edited to a high English standard. The particular reviewers’ comments I would like to draw your attention to are:

1. The abstract is too long and reviewer 2 provides a very useful, more precise version
2. Separate out the materials and methods from the introduction
3. Figures 1 and 2 can be kept but they need to be more professionally designed with titles without abbreviations, that do not repeat the figure number, minus a legend if only one dataset is presented and titles for the axes
4. Ensure that all values given through are consistent, particularly with regards to the number of participants
5. Address the experimental design critiques
6. Add a limitations section to the discussion and re-write the discussion to meet the concerns of reviewer 2

·

Basic reporting

1. Figures 1 and 2 are poorly annotated and formatted. Both are lacking a Y-axis title, and a proper figure legend. Figure 2 is also missing a figure heading.
2. The data represented in figures 1 and 2 can be easily summarized in a table. Using graphs to represent single percentage values does not seem necessary.
3. There are several errors throughout the manuscript in regards to the number of study subjects. Examples include: in line 40, the math is incorrect and the number of study subjects does not equal the total; also a math error in line 147.
4. There are also several spelling errors. Examples: line 101, use of the word “bellow” instead of “below”; line 201, use of the word “latter” instead of “later”

Experimental design

1. The statistical methods used to make each comparison should be more clearly described for each specific data table or figure. One possibility would be to write a description at the bottom of each table for the statistical method that was used to generate the p-values in that table.
2. Since antibiotic usage could influence H. pylori acquisition (line 215), and since there is high prevalence of antibiotic usage among HIV+ patients, the manuscript should report any data on antibiotic usage in this cohort.
3. The authors do not mention the reason for excluding patients with chronic diarrhea. Since diarrhea is a common symptom of HIV+ patients in both the pre and modern ART era, it is possible that many HIV+ patients were excluded unnecessarily.
4. The authors should state whether any of the patients are represented in multiple timepoint groups, for example if data was collected from an individual HIV+ patient during the pre-ART timepoint and the early or modern ART timepoint.

Validity of the findings

1. The conclusions are valid and supported by the data.

Additional comments

An alternative interpretation that should be considered is the direct impact of ART on H. pylori infection and gastritis, especially since there is evidence for ART-induced GI symptoms. (Wilcox CM, Saag MS Gastrointestinal complications of HIV infection: changing priorities in the HAART era. Gut 2008;57:861-870).

·

Basic reporting

The following layout has been followed:
1) Title
2) Abstract
3) Introduction
a) Materials and methods
b) Study inclusion criteria
c) Study exclusion criteria
d) Endoscopy and histology
e) Statistics
f) Study Approval
4) Results
5) Discussion

I would suggest Materials and Methods be a separate section header with the subheadings under this title rather than in the introduction.

1) Title
2) Abstract
3) Introduction
4) Materials and methods
a) Study inclusion and exclusion criteria
b) Endoscopy and histology
c) Statistics
d) Study Approval
5) Results
6) Discussion


There are numerous grammatical errors scattered throughout the text and it would be beneficial if a native English-speaking colleague review your manuscript.

The abstract contains unnecessary detail and should be contracted. It also needs to point out the basis for the conclusion that H pylori gastritis relates to immune reconstitution.

For example, consider the abstract rewritten thus:

Helicobacter pylori (H pylori) is one of the most common human bacterial infections with prevalence rates between 10-80% depending upon geographical location, age and socioeconomic status. H. pylori is commonly found in patients complaining of dyspepsia and is a common cause of gastritis.

During the course of their infection, people living with HIV (PLHIV) often have a variety of gastrointestinal symptoms including dyspepsia and while previous studies have reported HIV and H. pylori co-infection, there has been little data clarifying the factors influencing this.

The aim of this case-control study was to document the prevalence of H. pylori co-infection within the HIV community as well as to describe endoscopic findings, gastritis topography and histology, along with patient demographic characteristics across three different periods of time during which antiretroviral therapy (ART) has evolved, from pre- highly active antiretroviral therapy (HAART) to early and modern HAART eras. These data were compared to well-matched HIV negative controls.

Two hundred and sixteen PLHIV were compared with 1617 controls who underwent their first EGD to investigate dyspepsia. The prevalence of H.pylori co-infection among PLHIV was significantly higher in the early (32.1%) and modern HAART period (34.4%) compared with those with coinfection from the pre-HAART period (18.2%). The higher rates seen in patients from the HAART eras were similar to those observed among HIV negative controls (38.5%).

This prevalence increase among coinfected patients was in contrast to the fall in prevalence observed among controls, from 60.7% in the early period, to 52.9% in the second observed period. The three PLHIV co-infected subgroups differed regarding gastritis topography, morphology and pathology.

This study suggests that ART has an important impact on endoscopic and histological features of gastritis among HIV/H. pylori co-infected individuals, raising the possibility that H. pylori-induced gastritis could be an immune restoration disease.


Well referenced throughout.

Figures:
Generally helpful in the presentation of data.
It would be preferable to simplify the table headings and include a table description eg G1 = 1993-1997 etc.
Use SD not range
What does overall p value mean as a column given you are comparing three different groups? This needs to be explained.

Experimental design

This is a retrospective case:control study, which is level 3 evidence. This should be directly addressed in the limitations section of the discussion (which is currently absent).

Despite this limitation, the study represents original research with well-defined questions.

New data on temporal changes and putative relationship with ART, gastritis distribution and topography, risk factors for HIV and the relationship with H pylori infection.

Readily replicated study based on included description of design.

Validity of the findings

No justification provided for the selection of the date ranges. It is particularly unclear as to the date selection for the early-ART patient group.

Appropriate statistical analysis although it is not clear what the overall p value presented in the tables refers to.

In general, the discussion lets this paper down as there is little included that demonstrates the authors can interpret the data in a meaningful way for the reader. Rather, there is repetition of findings. For example, 225-234, there is inadequate interpretation of the findings and no attempt made to actually explain why this topography is similar between non-HIV patients and those in the post-ART era. In what way does the data provided by this study permit the following statement: All of these could suggest that the type and duration of immune and inflammatory responses are responsible for various histological presentations of H. pylori associated gastritis.

The statement: In HIV negative population, the low prevalence is due to the effective application of H. pylori eradication therapy, (line 213-14) is not based on data from this study but is presented as such and is unhelpful in interpreting the data presented.

215-221: This paper does not provide any insight into H pylori acquisition so it is difficult to comprehend why this is raised in this section of the paper.

Suggest removing the lines 222-224 as the inclusion/exclusion criteria were not validated in this study and consequently one cannot draw conclusions as to the effect of this on the study findings.

Additional comments

Thank you for your comprehensive report on HIV/H pylori coinfection. This study is well thought out and provides a significant amount of data to add to the collective knowledge-base of this challenging cohort of patients. Despite this, there are several areas of this paper that might benefit from revision to optimize the impact of the study. The most important aspect for review prior to consideration of publication would be the discussion where I feel the meaning and significance of your findings are not explained adequately. I also feel review of the entire paper by a native English speaker would substantially help in the flow of ideas and provide additional clarity around the data and its interpretation.

---

## Round 0.2 · accepted · Accept

You have addressed all of the comments made by the reviewers. There were still a few minor grammatical errors which I have corrected.